# CRIF1 siRNA-Encapsulated PLGA Nanoparticles Suppress Tumor Growth in MCF-7 Human Breast Cancer Cells

**DOI:** 10.3390/ijms24087453

**Published:** 2023-04-18

**Authors:** Shuyu Piao, Ikjun Lee, Seonhee Kim, Hyewon Park, Harsha Nagar, Su-Jeong Choi, Giang-Huong Vu, Minsoo Kim, Eun-Ok Lee, Byeong-Hwa Jeon, Dong Woon Kim, Youngduk Seo, Cuk-Seong Kim

**Affiliations:** 1Department of Physiology & Medical Science, School of Medicine, Chungnam National University, Daejeon 35015, Republic of Korea; 2Department of Anatomy and Cell Biology & Medical Science, School of Medicine, Chungnam National University, Daejeon 35015, Republic of Korea; 3Department of Nuclear Medicine, Chungnam National University Sejong Hospital, Sejong 30099, Republic of Korea

**Keywords:** CRIF1, MCF-7 cells, mitochondrial dysfunction, PLGA, nanoparticle

## Abstract

Mitochondrial oxidative phosphorylation (OXPHOS) system dysfunction in cancer cells has been exploited as a target for anti-cancer therapeutic intervention. The downregulation of CR6-interacting factor 1 (CRIF1), an essential mito-ribosomal factor, can impair mitochondrial function in various cell types. In this study, we investigated whether CRIF1 deficiency induced by siRNA and siRNA nanoparticles could suppress MCF-7 breast cancer growth and tumor development, respectively. Our results showed that CRIF1 silencing decreased the assembly of mitochondrial OXPHOS complexes I and II, which induced mitochondrial dysfunction, mitochondrial reactive oxygen species (ROS) production, mitochondrial membrane potential depolarization, and excessive mitochondrial fission. CRIF1 inhibition reduced p53-induced glycolysis and apoptosis regulator (TIGAR) expression, as well as NADPH synthesis, leading to additional increases in ROS production. The downregulation of CRIF1 suppressed cell proliferation and inhibited cell migration through the induction of G0/G1 phase cell cycle arrest in MCF-7 breast cancer cells. Similarly, the intratumoral injection of CRIF1 siRNA-encapsulated PLGA nanoparticles inhibited tumor growth, downregulated the assembly of mitochondrial OXPHOS complexes I and II, and induced the expression of cell cycle protein markers (p53, p21, and p16) in MCF-7 xenograft mice. Thus, the inhibition of mitochondrial OXPHOS protein synthesis through CRIF1 deletion destroyed mitochondrial function, leading to elevated ROS levels and inducing antitumor effects in MCF-7 cells.

## 1. Introduction

The mitochondria are the main sites of ATP production from carbon sources; they also perform many roles in tumorigenesis, including the regulation of cell signaling and cell death, generation of reactive oxidative species (ROS), and regulation of calcium homeostasis. Although the various roles of mitochondria in tumorigenesis have been extensively investigated, their involvement in cancer pathogenesis remains poorly under-stood. Cancer cells preferentially utilize glycolysis for ATP production, rather than mitochondrial oxidative phosphorylation (OXPHOS), because of the Warburg effect [1,2]. Therefore, researchers have placed considerable emphasis on the role of glucose metabolism in tumorigenesis. However, Warburg metabolism does not occur in all cancer cells; in some instances, cancer cells mainly depend on mitochondrial respiration to sustain the energy necessary for tumorigenesis. During the past decade, OXPHOS inhibition has been recognized as a potential target for cancer treatment [3,4,5]. Strategies to inhibit OXPHOS are typically categorized as inhibition of mitochondrial biogenesis, direct inhibition of respiratory chain complexes by various compounds, and indirect inhibition of OXPHOS via pharmacological effects. Metformin is a direct OXPHOS inhibitor that has repeatedly demonstrated antitumor effects in breast cancer and other cancers [6,7,8]. Numerous other compounds (e.g., BAY 87-2243, VLX600, and fenofibrate) have also been reported to directly inhibit OXPHOS [9,10,11,12]; several of these agents have demonstrated promising anti-cancer capacity and are undergoing phase I trials. However, there has been minimal re-search concerning the inhibition of mitochondrial protein synthesis as a treatment for cancer. Because tetracycline and doxycycline can prevent chronic myeloid leukemia and breast cancer [13,14,15], other inhibitors of mitochondrial protein synthesis should also be investigated for potential use in cancer therapy.

CR6-interacting factor 1 (CRIF1) is a mitochondrial ribosomal subunit protein that plays an important role in OXPHOS peptide synthesis in the inner membrane [16]. Previous studies have shown that CRIF1 downregulation causes misfolding of OXPHOS proteins, which leads to impaired OXPHOS protein expression in various cell lines [17,18,19]. We hypothesized that the deletion of CRIF1 would directly inhibit respiratory chain com-plex biogenesis, thus producing anti-tumor effects in MCF-7 breast cancer cells.

Small interfering RNA (siRNA)-based therapy has shown promising therapeutic effects in cancer treatment because of its ability to inhibit target gene mRNA expression [20,21]. However, because of the instability, erroneous biodistribution, and inefficient delivery of siRNA, there is a need to select efficient strategies for siRNA delivery into target cells. Poly (D, L-lactic-co-glycolic acid) (PLGA) is a copolymer that has been approved as a drug delivery carrier by the United States Food and Drug Administration. PLGA has demonstrated an effective delivery of siRNA into cells, through the inhibition of siRNA degradation, regulation of siRNA release time, and effective transport through lipid membranes [22]. In this study, we developed PLGA-encapsulated CRIF1 siRNA nanoparticles to deliver CRIF1 siRNA; we investigated whether CRIF1 downregulation shows antitumor effects in vitro and in vivo in an MCF-7 xenograft mouse model.

## 2. Results

### 2.1. CRIF1 Deficiency Disturbed Mitochondrial OXPHOS and Induced Mitochondrial Dysfunction

To determine the silencing efficiency of CRIF1 siRNA in breast cancer MCF-7 cells, we measured mRNA and protein expression levels of CRIF1 after siRNA transfection for 48 h. Treatment with CRIF1 siRNA induced efficient silencing of CRIF1 expression in MCF-7 cells (Figure 1A,B). Next, we investigated the effect of CRIF1 downregulation on mitochondrial function. Because CRIF1 has an essential role in the synthesis and integration of OXPHOS proteins, we measured the integrity of OXPHOS complexes in CRIF1-deficient MCF-7 cells. As expected, protein levels of the OXPHOS subunits NDUFA9 and UQCRC2 were lower in CRIF1-deficient cells than in siCon cells (Figure 1C). Consistent with these results, CRIF1 downregulation resulted in lower OCR values, less ATP synthesis, and reduced membrane potential, as well as a greater production of mitochondrial ROS (Figure 1D,E). We also observed mitochondrial morphology through cell staining with MitoTracker Red. The mitochondria of siCon cells exhibited a normal, thread-like shape, whereas the mitochondria of CRIF1-deficient cells exhibited a round, swollen shape, which indicated that CRIF1 deletion stimulated mitochondrial fission in MCF-7 cells. Overall, these findings suggest that CRIF1-deficient MCF-7 cells exhibit impaired OXPHOS and mitochondrial function.

### 2.2. CRIF1 Deficiency Reduced MCF-7 Cell Proliferation and Migration

CCK-8 assays were performed to determine the effects of CRIF1 downregulation on the proliferation of MCF-7 cells. Cells were transfected with CRIF1 siRNA; after 4 h of transfection, the serum-free medium was replaced with medium containing 25 mM glucose (hyperglycemic conditions) or 1 mM glucose (normoglycemic conditions) for another 48 h. Despite incubation with 25 mM glucose, CRIF1-deficient cells exhibited significantly less proliferation compared with the control cells; this effect was greater after incubation in 1 mM glucose. Proliferating cell nuclear antigen (PCNA) protein expression was also reduced in CRIF1-deficient cells. The effects of CRIF1 deficiency on cell migration were measured using wound-healing assays. CRIF1-deficient MCF-7 cells exhibited significantly less migration (by ~40%), compared with the control group (Figure 2C). Collectively, these results suggest that CRIF1 deletion-induced mitochondrial dysfunction led to reductions in the proliferation and migration of MCF-7 cells.

### 2.3. CRIF1 Deficiency Regulated Cell Cycle Progression but Did Not Affect Autophagy or Apoptosis in MCF-7 Cells

Because CRIF1 deletion is used as a target for mitochondrial dysfunction, we hypothesized that the impairment of mitochondrial function induces autophagy or apoptosis in MCF-7 cells. However, we observed that CRIF1 deficiency did not lead to the expression of autophagic markers (Beclin 1, p62, LCII) or apoptotic markers (poly [ADP-ribose] polymerase, Bcl-2, Bax) (Figure 3A). Additionally, p53, p16, and p21 participate in cell cycle regulation by arresting progression in the G0/G1 phase. The mRNA and protein levels of p53, p21, and p16 were significantly increased in CRIF1-deficient MCF-7 cells (Figure 3B). The percentage of cells in the G1 phase was substantially greater among CRIF1-deficient cells than among cells in the control group (Figure 3C). These results indicate that CRIF1 deficiency did not affect the induction of autophagic or apoptotic cell death, although it led to a greater percentage of MCF-7 cells in the G0/G1 phase.

### 2.4. CRIF1 Deficiency Inhibited Hypoxia-Induced Expression of HIF-1α and Led to Elevated Intracellular ROS Levels in MCF-7 Cells

Hypoxia is a characteristic environmental stress present in tumors. The expression of hypoxia-inducible factor 1α (HIF-1α) allows adaptation to the hypoxic environment, thus promoting cancer cell survival and progression. Mitochondrial dysfunction impairs hypoxia-induced expression of HIF-1α and delays tumor growth. Therefore, we examined whether CRIF1 deficiency-related changes in mitochondrial function also led to reduced expression of HIF-1α. We found that CRIF1 downregulation prevented hypoxia-induced accumulation of HIF-1α protein in MCF-7 cells (Figure 4A). TIGAR is strongly expressed in most breast carcinomas [23]; this strong expression promotes tumor growth through enhanced generation of NADPH-derived ROS. Therefore, we examined the expression of TIGAR in CRIF1-deficient cells, along with the levels of cytosolic ROS, which may be involved in cell cycle arrest. As expected, CRIF1 silencing reduced the mRNA and protein expression levels of TIGAR in MCF-7 cells (Figure 4B). These lower levels of TIGAR led to lower levels of NADPH through increases in the NADP/NADPH ratio (Figure 4C), thus contributing to a significant increase in intracellular ROS levels (Figure 4D). Together, these results demonstrate that CRIF1 deletion inhibited the hypoxia-induced expression of HIF-1α, and reduced the expression of TIGAR, resulting in higher intracellular ROS levels through the regulation of NADPH in MCF-7 cells.

### 2.5. CRIF1 siRNA-Encapsulated PLGA Nanoparticles Inhibited Tumor Growth in MCF-7 Xenograft Mice

To investigate the effects of CRIF1 siRNA in vivo, we used nanomaterials to transmit it into MCF-7 xenografts and examined whether PLGA-encapsulated CRIF1 siRNA nanoparticles inhibited the growth of MCF-7 xenografts by impairing mitochondrial function. We injected siCRIF1 nanoparticles into tumor tissue at 3 weeks after MCF-7 cell injection, then observed their antitumor effects (Figure 5A). The mean body weight did not significantly differ between the control and siCRIF1 nanoparticle groups throughout the experiment (Figure 5B). However, the mean tumor volume (128.8 mm3) after intratumoral injection was significantly lower in the siCRIF1 nanoparticle group than in the control group.

Tumor volume was also significantly lower in the siCRIF1 nanoparticle group (Figure 5D). Consistent with these findings, the siCRIF1 nanoparticle-treated group exhibited significantly lower tumor weight, compared with the siCon nanoparticle-treated group; there was no significant difference in body weight between the two groups (Figure 5E). To further investigate the antitumor mechanism of siCRIF1 nanoparticles in vivo, we analyzed the expression levels of proteins in the mitochondrial OXPHOS complex. The siCRIF1 nanoparticle-treated MCF-7 xenografts exhibited significant decreases in the expression levels of NDUFA9 and UQCRC2, compared with tumors from siCon-treated xenograft mice (Figure 6A). Next, we examined the expression levels of markers of cell proliferation (PCNA), metastasis (matrix metalloproteinase-9), and angiogenesis (CD31) through immunohistochemical staining. We observed significant reductions in the expression levels of PCNA, matrix metalloproteinase-9, and CD31 in the siCRIF1 nanoparticle group (Figure 6B). Hematoxylin and eosin staining was performed on tumor sections; the numbers of pyknotic nuclei were higher in the siCRIF1 nanoparticle-treated group than in the control group, indicating that cell death was induced in the siCRIF1 nanoparticle group. Next, we examined the expression of cell cycle-related proteins in tumors in vivo, in order to determine as to whether siCRIF1 nanoparticles exerted antitumor effects. The siCRIF1 nanoparticle injection group showed substantial increases in the expression levels of the cell cycle regulation proteins p53, p16, and p21, along with decreases in the expression levels of TIGAR and NADPH (Figure 6C). These results suggest that CRIF1 siRNA-encapsulated PLGA nanoparticles mediated antitumor effects in vivo by impairing the synthesis of OXPHOS subunits and inducing cell cycle arrest.

## 3. Discussion

Nanotechnology-based siRNA delivery systems offer a promising strategy for efficient and minimally toxic delivery of siRNA to target cells. In this study, we found that deletion of the mitochondrial ribosomal protein CRIF1 had an antitumor effect in MCF-7 breast cancer cells, which was mediated by G0/G1 phase arrest in vitro. Consistent with this finding, our newly developed CRIF1 siRNA-encapsulated nanoparticles also reduced tumor growth in MCF-7 xenograft mice in vivo.

In recent decades, increasing numbers of studies have demonstrated the potential of OXPHOS inhibitors, such as metformin and arsenic trioxide, as antitumor therapeutics. CRIF1 is a mitochondrial ribosome factor with essential roles in the synthesis and insertion of OXPHOS proteins into the inner membrane. CRIF1 deletion has been examined in multiple studies as a target for reducing the synthesis of mitochondrial OXPHOS proteins. Therefore, in this study, we investigated whether CRIF1 knockdown produced antitumor effects in breast cancer cells. Previous studies have demonstrated that CRIF1 is abundantly expressed in hepatocellular carcinoma, non-small cell lung cancer, and osteosarcoma [24,25,26]. CRIF1 deletion reportedly attenuates cell proliferation and metastasis; it also inhibits tumor growth in xenograft mice. Conversely, hepatic mitoribosomal defects induced by the disruption of CRIF1 expression aggravate liver cancer by regulating the immunometabolic microenvironment [27]. Additionally, CRIF1–CDK2 interface inhibitors decrease the resistance to anti-proliferative effects induced by paclitaxel in the MDA-MB-231 and MDA-MB468 breast cancer cell lines [28]. However, it has traditionally been unknown as to whether CRIF1 is overexpressed in MCF-7 breast cancer, or whether the CRIF1 deletion-induced disturbance of mitochondrial OXPHOS affects tumor progression in MCF-7 cells. In contrast to previous studies that observed a significant increase in CRIF1 expression in cancer cells, we observed no significant difference in CRIF1 expression between normal human mammary epithelial cells (MCF-10A) and breast cancer (MCF-7) cells (Appendix A). Therefore, we examined the effects of CRIF1 deficiency on mitochondrial OXPHOS subunit synthesis and function in MCF-7 cells and other cell lines; we found that CRIF1 deletion reduced the synthesis of mitochondrial OXPHOS proteins and induced severe mitochondrial dysfunction, which led to decreased ATP production, increased levels of mitochondrial ROS, and reductions in mitochondrial membrane potential and OCR. CRIF1 deficiency-induced mitochondrial dysfunction led to excessive ROS production and suppressed the proliferation and migration of MCF-7 cancer cells in vitro. Therefore, we conclude that methods targeting impaired mitochondrial OXPHOS protein synthesis represent promising strategies for breast cancer treatment. Although siRNA delivery remains a key challenge because of multiscale barriers, including the maintenance of effective and stable siRNA delivery in vivo, nanoparticles are able to overcome the limitations of alternative delivery systems [29]. In this study, we used PLGA nanoparticles as the delivery system because of their biocompatibility, time-controlled release, and safety profiles. The synthesized PLGA-encapsulated CRIF1 siRNA nanoparticles were administered by intratumoral injection, and CRIF1 gene-silencing was observed in tumor tissues (Figure 6A). The mean size of PLGA-encapsulated CRIF1 siRNA nanoparticles was 146.2 nm, and they sufficiently small to successfully enter MCF-7 cells according to our Zetasizer measurements (Appendix A) and scanning electron microscopy (Appendix A). Furthermore, the release assay indicated that nearly 93.23% of CRIF1 siRNA was released from the PLGA nanoparticles at 3 days (Appendix A).

The inhibition of mitochondrial OXPHOS induces apoptosis through the loss of mitochondrial membrane potential [30,31]. Elevated mitochondrial dysfunction also contributes to mitophagy, which eliminates damaged mitochondria and preserves cellular homeostasis. The activation of either apoptosis or mitophagy can lead to cancer cell death. However, in our study, we observed no changes regarding the induction of apoptosis or mitophagy in CRIF1-deficient MCF-7 cells, whereas a significant percentage of these cells exhibited G0/G1 phase arrest during the cell cycle. The disruption of mitochondrial homeostasis (e.g., mitochondrial stress, dynamics, and function) mediates deceleration of the cell cycle, in conjunction with elevated levels of ROS and DNA damage [32,33,34]. Therefore, CRIF1 deficiency destroyed mitochondrial homeostasis and caused a significant inhibition of cell growth. The reduced expression of TIGAR also led to the inhibition of antioxidant NADPH production, which increased the production of ROS, and may have contributed to cell cycle regulation.

In conclusion, we demonstrated that CRIF1 downregulation had antitumor effects in MCF-7 cells through the inhibition of cell proliferation and migration in vitro. We also developed PLGA-encapsulated siRNA nanoparticles and confirmed their antitumor effects in MCF-7 xenograft mice in vivo. The results garnered in this study constitute pre-clinical in vivo evidence of the beneficial effects of siCRIF1 nanoparticles in the treatment of MCF-7 breast cancer; the findings will facilitate the development of siCRIF1 nanoparticle-based therapeutics.

## 4. Materials and Methods

### 4.1. Cell Culture and Transfection

Breast cancer cells (MCF-7) (ATCC HTB 22) were purchased from the American Type Culture Collection and cultured in Minimum Essential Medium, supplemented with 10% fetal bovine serum, 1% antibiotics, sodium pyruvate, non-essential amino acids, and insulin; incubation was conducted at 37 °C with 5% CO2. MCF-7 cells were transfected with negative control siRNA and siRNA targeting CRIF1 (human siRNA sequence: sense, 5′-UGGAGGCCGAAGA ACGCGAAUGGUA-3′ and antisense, 5′-UACCAUUCGCGUUCU UCGGCCUCCA-3′) using the Lipofector-2000 (AB-LF-2002, AptaBio, Seoul, Republic of Korea). After siRNA transfection for 48 h, CRIF1 silencing efficiency was assessed through Western blotting analysis and quantitative reverse-transcription polymerase chain reaction (qRT-PCR).

### 4.2. Establishment of Subcutaneous Xenografts in Nude Mice

Four-week-old female BALB/c nude mice purchased from OrientBio (Seoul, Republic of Korea) were housed under specific pathogen-free conditions with ad libitum access to a γ-ray-irradiated laboratory rodent diet (Purina Korea, Seongnam, Republic of Korea) and autoclaved water; the housing temperature was 24 °C, and a 12-h/12-h day/night light cycle was used. All animal procedures were performed in accordance with the guidelines and regulations of the animal care unit of the Chungnam National University Institutional Animal Care and Use Committee. Before the establishment of xenografts in nude mice, a 1.7 mg/60-day release 17β-estradiol pellet (Innovative Research of America, Sarasota, FL, USA) was implanted into the intrascapular space of each mouse for 2 weeks. Xenografts were created in 7-week-old mice through the injection of MCF-7 cells. After each mouse had been anesthetized, MCF-7 cells (5 × 106 cells/mouse; 1:1 ratio with Matrigel) suspended in a volume of 100 μL were subcutaneously injected into the right side of the lower limb of the mouse. Body weight and tumor size were measured periodically. Tumor volumes were measured using a caliper and estimated as follows: tumor volume = (length × width2) × π/6. At 3 weeks after the injection of MCF-7 cells, 20 μL of solution (siCon or siCRIF1 PLGA-encapsulated nanoparticles) was injected into the tumor using a syringe with a 26-gauge needle. Five weeks after the injection of nanoparticles, tumor tissues were harvested and used for subsequent analysis.

### 4.3. Oxygen Consumption Rate (OCR) Measurements

OCR values were measured using a Seahorse XF-24 analyzer (Seahorse Biosciences, North Billerica, MA, USA). On the day before measurement, the sensor cartridge was calibrated at 37 °C. After cells had been transfected with siCon or siCRIF1 for 24 h, they were harvested and seeded in Seahorse XF-24 plates at a density of 30,000 cells/well, then incubated for another 24 h. On the day of OCR measurement, cells were washed twice with XF assay media and incubated in a 37 °C incubator without CO_2_ for 1 h. Three readings were acquired after each addition of mitochondrial inhibitor, prior to the injection of electron transport inhibitors (2 μg/mL oligomycin, 0.5 μM carbonyl cyanide m-chlorophenyl hydrazone, and 2 μM rotenone). OCR values were automatically measured by the sensor cartridge and recorded by the Seahorse XF-24 software.

### 4.4. Flow Cytometry-Based Analysis of Cell Cycle

After siCRIF1 transfection for 48 h, cells were washed twice with phosphate-buffered saline (PBS); they were then harvested by trypsinization and fixed with cold 70% ethanol at 4 °C overnight. The fixed cells were washed twice with cold PBS and incubated in PBS containing 5 μg/mL propidium iodide (cat. no. 00-6990-50, Thermo Fisher Scientific, Waltham, MA, USA) and 1 mg/mL RNaseA (R875-100 mg, Sigma-Aldrich, St. Louis, MO, USA) for 30 min at 37 °C in the dark. Cell cycle analyses were immediately performed using a NovoCyte flow cytometer (Agilent Technologies, Santa Clara, CA, USA) and NovoExpress 1.4.1 software (ACEA Bioscience Inc., San Diego, CA, USA).

### 4.5. Immunoblotting

Tumor tissues or cells were lysed with radioimmunoprecipitation assay lysis buffer, containing a protease and phosphatase inhibitor cocktail. After the lysates had been centrifuged at 12,000× *g* rpm for 15 min, the protein concentration of the supernatant was determined using a bicinchoninic acid protein assay kit (iNtRON Biotechnology Co., Seongnam, Republic of Korea). Equal amounts of protein were separated by sodium dodecyl sulfate polyacrylamide gel electrophoresis and transferred onto polyvinylidene difluoride membranes (Immobilon-PSQ, Millipore, Tullagreen, Ireland). The membranes were blocked with 5% skim milk for 1 h at room temperature, then incubated with primary antibodies for p16, p21, p53, and Tigar (Santa Cruz Biotechnology, Santa Cruz, CA, USA). Primary antibodies for the OXPHOS complex subunits NDUFA9, SDHA, UQCRC2, and ATP5a1 were purchased from Invitrogen (Carlsbad, CA, USA). We also used primary antibodies for β-actin and PCNA (Sigma-Aldrich), COX-4 (Cell Signaling Technology, Beverly, MA, USA), α-tubulin (R & D Systems, Minneapolis, MN, USA), HIF-1α (Cayman Chemical, Michigan, MI, USA) and CRIF1 (Abcam, Cambridge, UK). After membranes had been washed 3 times in Tris-buffered saline plus Tween for 10 min each, they were incubated with appropriate peroxidase-conjugated secondary antibodies for 1 h at room temperature, then washed 3 times in Tris-buffered saline plus Tween for 10 min each. Chemiluminescent signals were detected using Super Signal West Pico or Femto Substrate (Thermo Fisher Scientific, Waltham MA, USA).

### 4.6. NADP/NADPH Measurements

After the treatment as described above, NADP/NADPH levels were detected in MCF-7 cells using a NADP/NADPH quantification colorimetric kit (BioVision, Miltipas, CA, USA), in accordance with the manufacturer’s instructions. Briefly, cells were lysed with NADP+/NADPH extraction buffer and then centrifuged. Supernatants were collected and subjected to measurements using a microplate reader at 450 nm.

### 4.7. Immunofluorescence

To detect changes in mitochondrial morphology, cells were grown on glass coverslips, then transfected with negative control siRNA or CRIF1 siRNA for 48 h. After cells had been washed with PBS, they were stained with MitoTracker Red FM (Cell Signaling Technology, Topsfield, MA, USA) at 37 °C for 30 min in the dark. Subsequently, cells were washed with Hank’s balanced salt solution and mounted with Vectashield anti-fade mounting medium containing 4′, 6-diamidino-2-phenylindole (DAPI) (Vector Labs, Peterborough, UK). Images were obtained using a confocal laser microscope (LSM700, Carl Zeiss, Jena, Germany).

### 4.8. CCK-8 Proliferation Assay

Cell proliferation was measured using a CCK-8 kit (Dojindo, Tokyo, Japan), in accordance with the manufacturer’s instructions. Briefly, cells seeded in 6-well plates were washed with PBS, then incubated in 10% CCK-8 diluted in growth media. These cells were incubated at 37 °C for 1 h in the dark; the absorbance was measured at 450 nm using a microplate reader.

### 4.9. Measurements of Mitochondrial Membrane Potential and Mitochondrial ROS Production

After 48 h of transfection with negative control siRNA or CRIF1 siRNA, cells were washed twice with PBS and trypsinized. For measurements of mitochondrial membrane potential, cells were incubated with tetramethyl rhodamine ethyl ester dye (Invitrogen, Carlsbad, CA, USA) at 37 °C for 15 min in the dark. For measurements of mitochondrial ROS production, cells were incubated with 3 μM MitoSOX red fluorescence (Invitrogen, Waltham, MA, USA) at 37 °C for 15 min in the dark. After the incubation period, each sample was washed twice with PBS and examined using the Fluoroskan Ascent fluorescence reader (Thermo Fisher Scientific, Waltham, MA, USA) at excitation and emission wavelengths of 530 and 590 nm, respectively.

### 4.10. Wound-Healing Assay

For the wound-healing assay, MCF-7 cells were transfected with negative control siRNA or CRIF1 siRNA in 6-well dishes for 24 h to reach 100% confluence. Next, the monolayers were scratched with a sterile 200 μL pipette tip to detach cells from the center of the well. Floating cells were washed twice with PBS to remove cell debris. Then, the plates were replaced with serum-free medium and incubated for another 24 h; the total incubation time after transfection was 48 h. Images were acquired at 24 h after scratching, and the migration area was measured using ImageJ software (version 1.46, National Institutes of Health, Bethesda, MD, USA).

### 4.11. qRT-PCR Analysis

Total RNA was isolated using TRIzol Reagent (Invitrogen). Total RNA concentrations were measured using a C40 nanophotometer (Implen, Munich, Germany). Complementary DNA (cDNA) was prepared using the Maxime RT Premix kit (iNtRON Biotechnology). Relative RNA levels were determined using the Prism 7000 Sequence Detection System (Applied Biosystems, Foster City, CA, USA), with the Super Script III Platinum SYBR GreenOne-Step qRT-PCR Kit (Invitrogen). The primer sequences for amplification of human TIGAR, p16, p21, p53 were as follows: TIGAR-sense-5′-CCTTACCAGCCACTCTGAGC-3′ and antisense-5′-CCATGTGCAATCCAGAGATG-3′; p16-sense-5′-GTGAGGGTTTTCGTGGT TCAC-3′ and antisense-5′-CTGGTCTTCTAGGAAGCGGCT-3′; p21-sense-5′-GGACAGCA GAGGAAGACCATG-3′ and antisense- 5′-CGGCGTTTGGAGTGGTAGAA-3′; p53-sense-5′-GAGGTTGGCTCTGACTGTACC-3′ and antisense- 5′-TCCGTCCCAGTAGATTACCA C-3′. The cycle conditions were as follows: 5 min at 95 °C; 40 cycles of 30 s at 95 °C, 30 s at 60 °C, and 30 s at 72 °C; and 5 min at 72 °C. Dissociation curves were calculated to identify aberrant primer dimer formation. The 2−ΔΔCt method was used to determine fold changes in gene expression.

### 4.12. Preparation of PLGA-Encapsulated CRIF1 siRNA Nanoparticles

PLGA-encapsulated CRIF1 siRNA nanoparticles were prepared by Nanoglia Inc. (Daejeon, Republic of Korea). First, 20 μM CRIF1 siRNA in 200 μL Tris–ethylenediaminetetraacetic acid buffer (pH 8.0) was added dropwise to 800 µL of dichloromethane, containing 25 mg of PLGA (Corbion, Amsterdam, Netherlands); the mixture was emulsified by sonication (50 W, 1 min; Vibra-Cell VCX 130, Sonics, Newtown, CT, USA) into a W1/O emulsion. The emulsion was diluted with 5 mL of 1% PVA1500 (*w*/*v*), then stirred with a magnetic stirring bar for 3 h at room temperature to evaporate the dichloromethane. Finally, the PLGA nanoparticles were collected by ultracentrifugation at 15,000× *g* for 15 min at 4 °C, then washed with deionized water, and freeze-dried for 24 h. The physical characteristics of the nanoparticles were analyzed using a Zetasizer Nano ZS instrument (Malvern Instruments, Malvern, UK) and an electron microscope. The release of CRIF1 siRNA from the nanoparticles was assessed by incubation for 0, 1, 2, or 3 days. The supernatant was harvested and diluted with PBS, and the percentage of released CRIF1 siRNA was measured using a NanoDrop (Thermo Fisher Scientific, Waltham MA, USA).

### 4.13. Statistical Analyses

Statistical analyses were performed using GraphPad Prism 8.0 software (GraphPad Software, San Diego, CA, USA). Data are presented as means ± standard errors of the mean. Differences between two groups were evaluated using two-tailed Student’s *t*-tests. Multiple comparisons were performed using one-way analysis of variance, followed by Tukey’s post hoc test. *p*-values < 0.05 were considered statistically significant. Data are representative of ≥3 independent experiments.

## Figures and Tables

**Figure 1 ijms-24-07453-f001:**
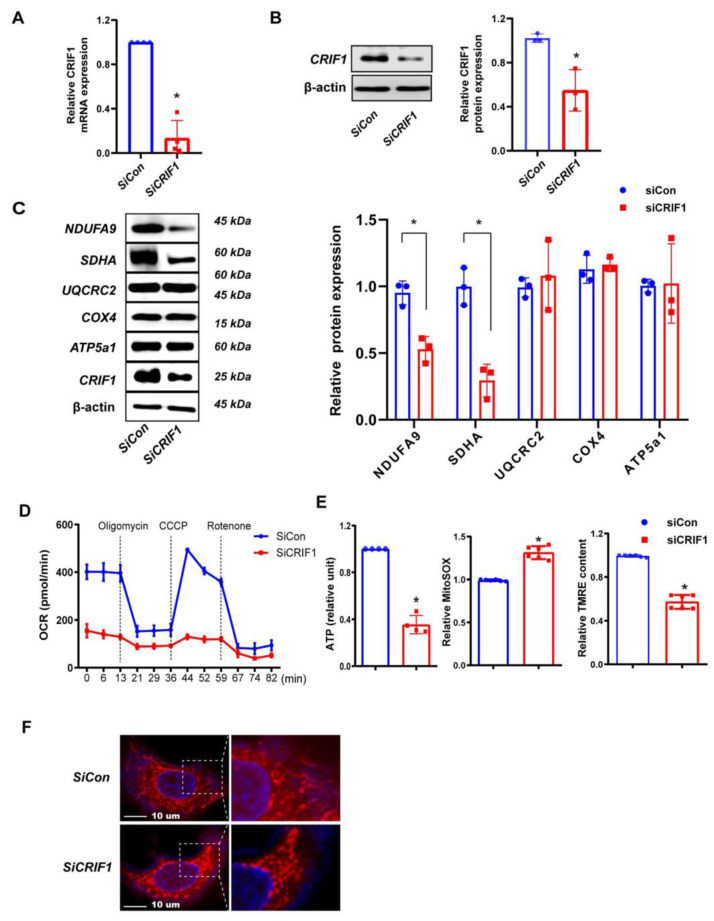
CRIF1 deficiency disturbed mitochondrial OXPHOS and induced mitochondrial dysfunction in MCF-7 cells. MCF-7 cells were transfected with 100 pmol of siCon and siCRIF1 for 48 h. (**A**) CRIF1 mRNA levels was determined by qPCR. (**B**) CRIF1 protein levels and (**C**) mitochondrial OXPHOS complex subunits were detected by immunoblotting and then quantified by densitometric analysis. (**D**) OCR was analyzed by a Seahorse XF-24 flux analyzer. (**E**) Relative ATP levels, MitoSOX, and TMRE were measured by using a colorimetric/fluorometric assay kit. (**F**) Representative staining images of Mitotracker-red and DAPI. High-magnification images from the selected area are shown in the right. Scale: 10 μm. All data are presented as means ± SEM of at least three independent experiments. * *p* < 0.05 compared with siCon group.

**Figure 2 ijms-24-07453-f002:**
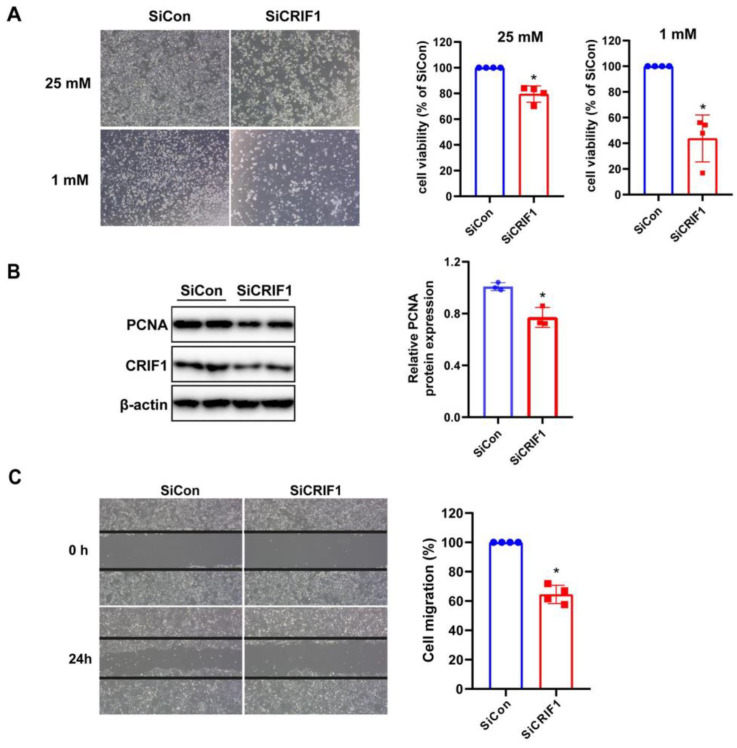
CRIF1 deficiency reduced MCF-7 cell proliferation and migration. MCF-7 cells were transfected with 100 pmol of siCon and siCRIF1 for 48 h. (**A**) Morphology of MCF-7 cells grown in media containing 25 mM or 1 mM glucose. Magnification, ×100. (**B**) PCNA protein level was detected by Western blotting. β-actin was used as a loading control. Protein levels were quantified by densitometric analysis. (**C**) Cell migration was detected using the cell scratch assay. Magnification, ×40. All data are presented as means ± SEM of at least three independent experiments. * *p* < 0.05 compared with siCon group.

**Figure 3 ijms-24-07453-f003:**
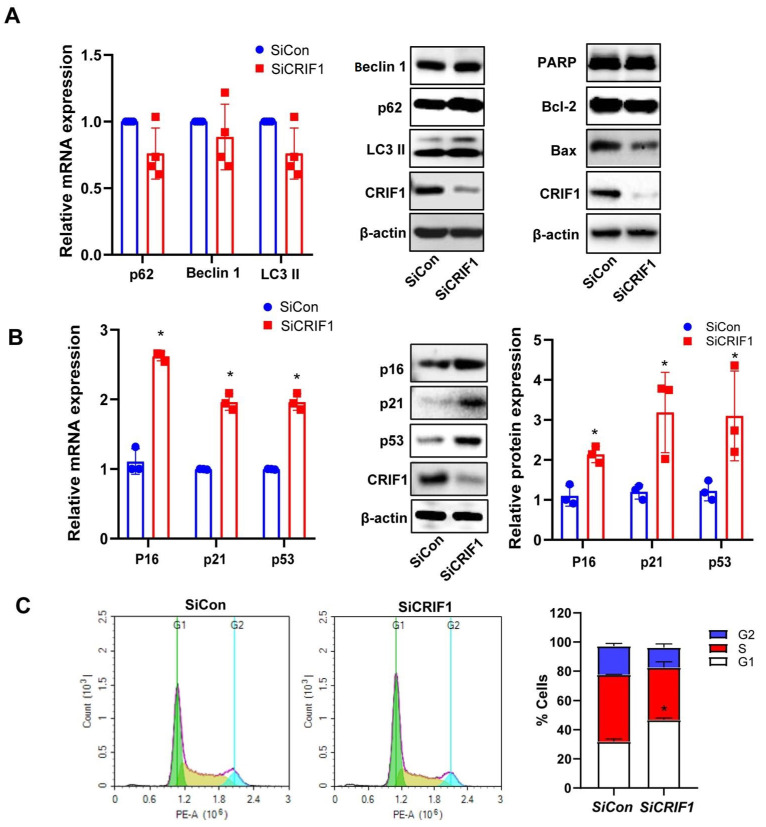
CRIF1 deficiency regulated cell cycle progression but did not affect autophagy or apoptosis in MCF-7 cells. MCF-7 cells were transfected with 100 pmol of siCon and siCRIF1 for 48 h. (**A**) mRNA expressions of p62, beclin1, LC3II were detected by qPCR. Autophagy and apoptosis-related proteins were detected by immunoblotting. (**B**) mRNA expressions of p16, p21, p53 were detected by qPCR. p16, p21, and p53 protein expression were detected by immunoblotting and then quantified by densitometric analysis. (**C**) Cell cycle status was detected by using FACS analysis. Green shows G1 phase, yellow shows S phase, and blue shows G2 phase. All data are presented as means ± SEM of at least three independent experiments. * *p* < 0.05 compared with siCon group.

**Figure 4 ijms-24-07453-f004:**
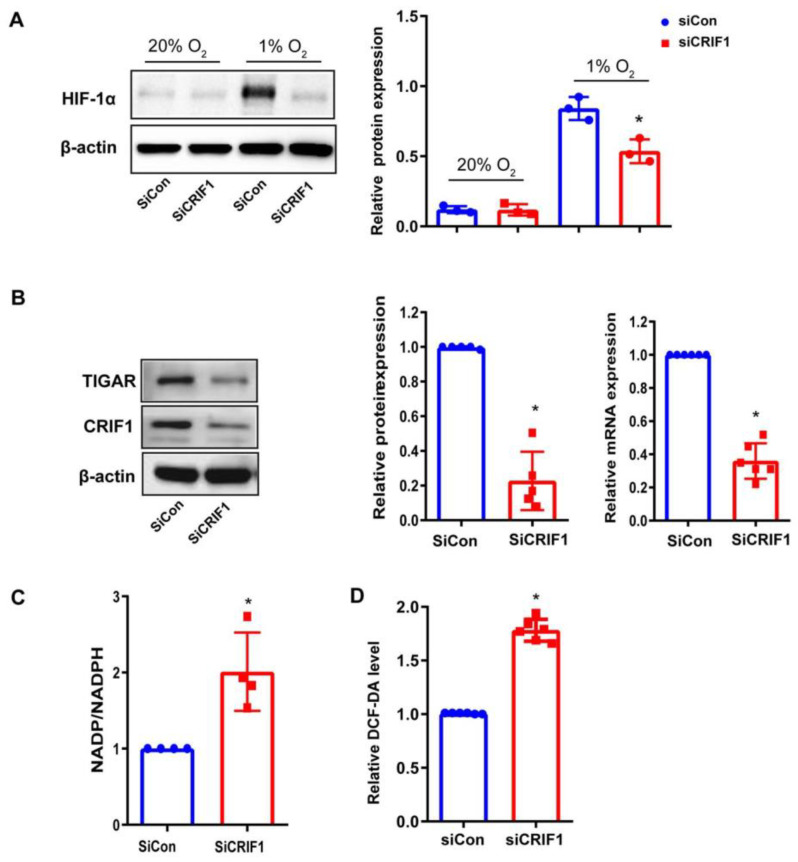
CRIF1 deficiency inhibited hypoxia-induced expression of HIF-1α and led to elevated intracellular ROS levels in MCF-7 cells. (**A**) MCF-7 cells were transfected with 100 pmol of siCon and siCRIF1 and then incubated in 20% and 1% O2 incubator for 48 h, respectively. Western blot analysis of HIF-1α (**B**) mRNA and protein expression of TIGAR were measured by qPCR and immunoblotting, respectively. (**C**) Total NADP/NADPH ratio and (**D**) DCF-DA fluorescence intensity were measured by using a colorimetric/fluorometric assay kit. All data are presented as means ± SEM of at least three independent experiments. * *p* < 0.05 compared with siCon group.

**Figure 5 ijms-24-07453-f005:**
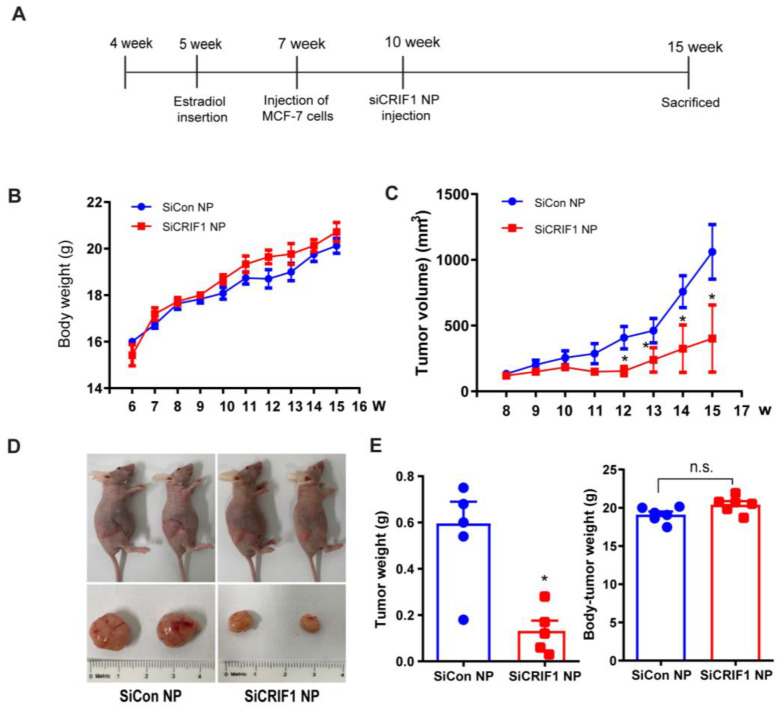
CRIF1 siRNA-encapsulated PLGA nanoparticles inhibited tumor growth in MCF-7 xenograft mice. (**A**) Schematic representation of MCF-7 xenograft mouse model and administration of NP. (**B**) Body weight. (**C**) Tumor volume. (**D**) Representative photographs of tumor-bearing mice and extracted tumor. (**E**) Tumor weight and body-tumor weight in siCon NP and siCRIF1 NP groups. All data are presented as means ± SEM of at least three independent experiments (*n* = 5 mice in each group; * *p* < 0.05 vs. siCon NP group).

**Figure 6 ijms-24-07453-f006:**
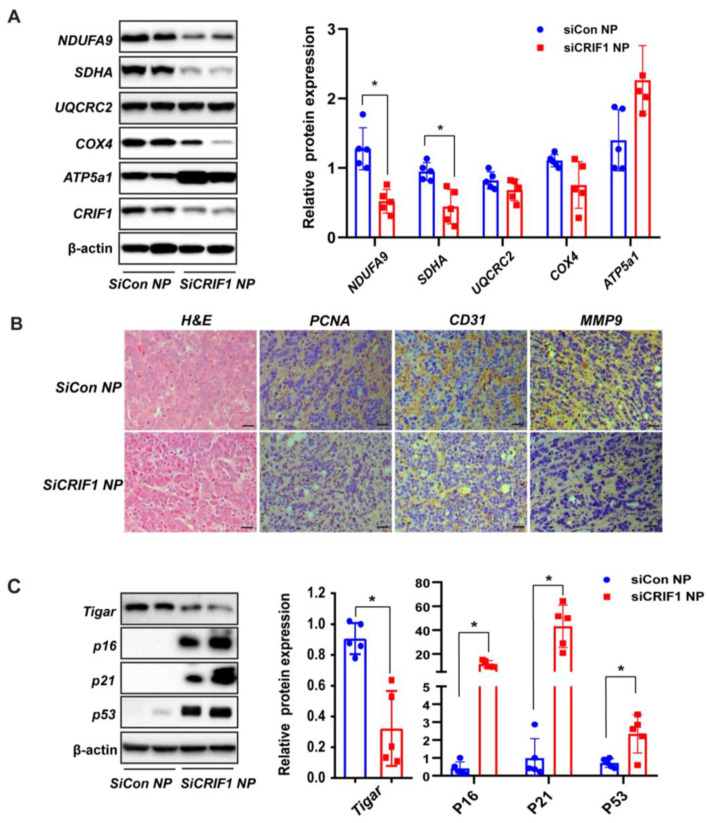
CRIF1 siRNA-encapsulated PLGA nanoparticles impaired mitochondrial function and suppressed the cell cycle. (**A**) mitochondrial OXPHOS complex subunits in tumor tissues were detected by immunoblotting and then quantified by densitometric analysis. (**B**) Isolated tumors were stained with H&E, PCNA, CD31, MMP9. Scale bar, 50 μm. (**C**) TIGAR, p16, p21, and p53 protein expressions were detected by Western blotting and then quantified by densitometric analysis. All data are presented as means ± SEM of at least three independent experiments (*n* = 5 mice in each group; * *p* < 0.05 vs. siCon NP group).

## Data Availability

Data is contained within the article.

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
