# Peer review of "CRIF1 siRNA-Encapsulated PLGA Nanoparticles Suppress Tumor Growth in MCF-7 Human Breast Cancer Cells"

_ijms, 2023, doi:10.3390/ijms24087453_

Round 1
Reviewer 1 Report
The authors focused on CRIF1 deletion which used as a target for mitochondrial dysfunction. They designed well and performed any required in vitro/in vivo experiments. They hypothesized that the impairment of mitochondrial function induces autophagy or apoptosis in MCF-7 cells. They observed that CRIF1 deficiency did not lead to the expression of autophagic markers (Beclin 1, p62, LCII) or apoptotic markers (poly [ADP-ribose] polymerase, Bcl-2, Bax). Only cell cycle regulators such a p53, p16, and p21 were involved.
It would be accepted if minor methodological errors and text editing were solved by any experts.
Reviewer 2 Report
Overall, this is a clear and well-written manuscript. This study has some clinical significance. Nonetheless, the manuscript can be further improved and the following concerns should be adequately addressed. My detailed comments are as follows
1. In Line 7, the author needs to list the right e-mail address.
2. In Line 9, the author needs to list the right e-mail address.
3. The author needs to add some results about PLGA nanoparticles, such SEM, and diameter.
4. How is siRNA released from PLGA nanoparticles? The author needs to investigate the siRNA release curve.
